# Revealing computational mechanisms of retinal prediction via model reduction

**Hidenori Tanaka[1], Aran Nayebi[3], Niru Maheswaranathan[3,4], Lane McIntosh[3], Stephen A. Baccus[2], and Surya Ganguli[1,4]**

[1]Department of Applied Physics, Stanford University, Stanford, CA 94305
[2]Department of Neurobiology, Stanford University, Stanford, CA 94305
[3]Neurosciences PhD Program, Stanford University, Stanford, CA 94305
[4]Google Brain, Google, Inc., Mountain View, CA 94043

## Abstract

Recently, deep feedforward neural networks have achieved considerable success in modeling biological sensory processing, in terms of reproducing the input-output map of sensory neurons. However, such models raise profound questions about the very nature of explanation in neuroscience. Are we simply replacing one complex system (a biological circuit) with another (a deep network), without understanding either? Moreover, beyond neural representations, are the deep network's *computational mechanisms* for generating neural responses the same as those in the brain? Without a systematic approach to extracting and understanding computational mechanisms from deep neural network models, it can be difficult both to assess the degree of utility of deep learning approaches in neuroscience, and to extract experimentally testable hypotheses from deep networks. We develop such a systematic approach by combining dimensionality reduction and modern attribution methods for determining the relative importance of interneurons for specific visual computations. We apply this approach to deep network models of the retina, revealing a conceptual understanding of how the retina acts as a predictive feature extractor that signals deviations from expectations for diverse spatiotemporal stimuli. For each stimulus, our extracted computational mechanisms are consistent with prior scientific literature, and in one case yields a new mechanistic hypothesis. Thus overall, this work not only yields insights into the computational mechanisms underlying the striking predictive capabilities of the retina, but also places the framework of deep networks as neuroscientific models on firmer theoretical foundations, by providing a new roadmap to go beyond comparing neural representations to extracting and understand computational mechanisms.

## 1 Introduction

Deep convolutional neural networks (CNNs) have emerged as state of the art models of a variety of visual brain regions in sensory neuroscience, including the retina [1, 2] and deeper visual areas. Their success has so far been primarily evaluated by their ability to explain reasonably large fractions of variance in biological neural responses across diverse visual stimuli. However, fraction of variance explained is not of course the same thing as scientific explanation, as we may simply be replacing one inscrutable black box (the brain), with another (a potentially overparameterized deep network).

Indeed, any successful scientific model of a biological circuit should succeed along three fundamental axes, each of which goes above and beyond the simple metric of mimicking the circuit's input-output map. First, the intermediate *computational mechanisms* used by the hidden layers to generate

33rd Conference on Neural Information Processing Systems (NeurIPS 2019), Vancouver, Canada.

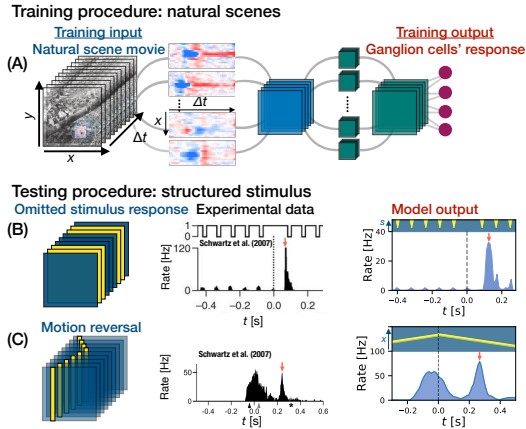

Training procedure: natural scenes

Testing procedure: structured stimulus

Figure 1: **Deep learning models of the retina trained only on natural scenes reproduce predictive retinal phenomena [2].**
(A) Training procedure: We analyzed a three-layer convolutional neural network (CNN) model of the retina which takes as input a spatiotemporal natural scene movie and outputs a nonnegative firing rate, corresponding to a retinal ganglion cell response. The first layer consists of eight spatiotemporal convolutional filters (i.e., cell types), the second layer of eight convolutional filters, and the fully connected layer predicting the ganglion cells' response. The deep learning model reproduces [2] (B) an omitted stimulus response [5], and (C) the motion reversal response [6].

responses should ideally match the intermediate computations in the brain. Second, we should be able to extract *conceptual insight* into *how* the neural circuit generates nontrivial responses to interesting stimuli. And third, such insights should suggest new experimentally testable hypotheses that can drive the next generation of neuroscience experiments.

However, it has been traditionally difficult to algorithmically extract computational mechanisms, and consequently conceptual insights, from deep CNN models due to their considerable complexity. Here we provide a method to do so based on the idea of model reduction, whose goal is to systematically extract a simple, reduced, minimal subnetwork that is most important in generating a complex CNN's response to any given stimulus. Such subnetworks then both summarize computational mechanisms and yield conceptual insights. We build on ideas from interpretable machine learning, notably methods of input attribution that can decompose a neural response into a sum of contributions either from individual pixels [3] or hidden neurons [4]. To achieve considerable model reduction for responses to spatiotemporal stimuli, we augment and combine such input attribution methods along with dimensionality reduction.

## 2    From deep CNNs to neural mechanisms through model reduction

To extract understandable reduced models from the millions of parameters comprising the deep CNN in Fig. 1A and [2], we first carve out important sub-circuits using modern attribution methods [3, 4], and then reduce dimensionality by exploiting spatial invariances present in the artificial stimuli carefully designed to specifically probe retinal physiology (Fig.1B-C). We proceed in 3 steps:

**Step (1): Quantify the importance of a model unit.**

Since one main goal here is model reduction, we consider attributing the ganglion cell response back to the first layer of hidden units to quantify their importance. We denote by $z_{cxy}^{[1]}(t) = W_{cxy}^{[1]} \circledast s(t) + b_{cxy}$ the pre-nonlinearity activation of the first layer (denoted by [1]) hidden units, where $W_{cxy}$ and $b_{cxy}$ are the convolutional filters and biases of a unit in channel $c$ ($c = 1, \ldots, 8$) at convolutional position $(x, y)$ (with $x, y = 1, \ldots, 36$), and $s(t)$ is the spatiotemporal input stimulus. Now computing the line integral $\mathcal{F}[s(t; 1)] = \int_0^1 d\alpha \left. \frac{\partial \mathcal{F}}{\partial z^{[1]}} \right|_{s(t,\alpha)} \cdot \frac{\partial z^{[1]}}{\partial \alpha}$ over a straight path in spatiotemporal stimulus space from the zero stimulus to $s(t)$ given by $s(t; \alpha) = \alpha s(t)$ where the path parameter $\alpha$ ranges from 0 to 1 yields

$$r(t) = \sum_{x,y,c} \left[ \int_0^1 d\alpha \left. \frac{\partial \mathcal{F}}{\partial z_{cxy}^{[1]}} \right|_{s(t,\alpha)} \right] (W_{cxy}^{[1]} \circledast s) = \sum_{x,y,c} [\mathcal{G}_{cxy}(s)] (W_{cxy}^{[1]} \circledast s) = \sum_{x,y,c} \mathcal{A}_{cxy}. \quad (1)$$

This represents an *exact* decomposition of the response $r(t)$ into attributions $\mathcal{A}_{cxy}$ from each subunit at the same time $t$ (since all CNN filters beyond the first layer are purely spatial). This attribution further splits into a product of $W_{cxy}^{[1]} \circledast s$, reflecting the activity of that subunit originating from spatiotemporal filtering of the preceding stimulus history, and an effective stimulus dependent weight $\mathcal{G}_{cxy}(s)$ from each subunit to the ganglion cell, reflecting how variations in subunit activity $z_{cxy}^{[1]}$

as the stimulus is turned on from 0 to $s(t)$ yield a net impact on the response $r(t)$. A positive (negative) effective weight indicates that increasing subunit activity along the stimulus path yields a net excitatory (inhibitory) effect on $r(t)$.

**Step (2): Exploiting stimulus invariances to reduce dimensionality.** The attribution of the response $r(t)$ to first layer subunits in (1) still involves $8 \times 36 \times 36 = 10,368$ attributions. We can, however, leverage the spatial uniformity of artificial stimuli used in neurophysiology experiments to reduce this dimensionality. For example, in the omitted stimulus response (OSR), stimuli is spatially uniform, implying $W_c^{[1]} \circledast s \equiv W_{cxy}^{[1]} \circledast s$ is independent of spatial indices $(x, y)$. Thus, we can reduce the number of attributions to the number of channels via

$$r(t) = \sum_{c=1}^{8} \left( \sum_{x=1}^{36} \sum_{y=1}^{36} \mathcal{G}_{cxy}(s) \right) \cdot (W_c^{[1]} \circledast s) \equiv \sum_{c=1}^{8} \mathcal{G}_c(s) \cdot (W_c^{[1]} \circledast s) \equiv \sum_{c=1}^{8} \mathcal{A}_c. \tag{2}$$

More generally for other stimuli with no obvious spatial invariances, one could still attempt to reduce dimensionality by performing PCA or other dimensionality reduction methods on the space of hidden unit pre-activations or attributions over time. We leave this intriguing direction for future work.

**Step (3): Building reduced models from important subunits.** Finally, we can construct minimal circuit models by first identifying "important" units defined as those with large magnitude attributions $\mathcal{A}$. Second, we construct our model as a one hidden layer neural network composed of only the important hidden units, with effective connectivity from each hidden unit to the ganglion cell determined by the effective weights $\mathcal{G}$ in (1), or (2).

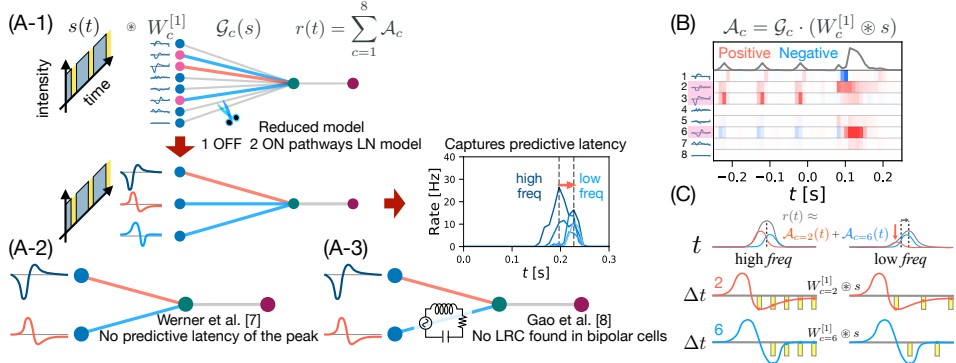

Figure 2: **Omitted stimulus response (OSR)** (A-1) Schematics of the model reduction procedure by only leaving three (1 OFF, 2 ON) highly contributing units. (A-2,3) Previously proposed models of the OSR phenomenon [7, 8]. (B) Attribution for each of the cell types $\mathcal{A}_c$ over time. (C) Two ON bipolar cells are necessary to capture the predictive latency. Cell 2 with earlier peak is only active in a high-frequency regime, while the cell 6 with later peak is active independent of the frequency.

We apply our attribution method to identify the subcircuit responsible for the omitted stimulus response (OSR) phenomenon (Fig. 1B), in which a periodic sequence of full field flashes entrains a retinal ganglion cell to respond periodically, but when a single flash is omitted, the ganglion cell produces an even larger response at the expected time of the response to the omitted flash. This OSR phenomenon is observed across several species including salamander [5, 9]. Interestingly, for periodic flashes in the range of 6-12Hz, the latency between the last flash before the omitted one, and the burst peak in the response, is proportional to the period of the train of flashes [5, 9], indicating the retina retains a short memory trace of this period. Moreover, pharmacological experiments suggest ON bipolar cells are required to produce the OSR [7, 9], which have been shown to correspond to the first layer hidden units in the deep CNN [1, 2].

These phenomena raise two fundamental questions: what computational mechanism causes the large amplitude burst, and how is the timing of the peak sensitive to the period of the flashes? There are two theoretical models in the literature that aim to answer these questions. One proposes that the bipolar cell activity responds to each individual flash with an oscillatory response whose period

adapts to the period of the flash train [8]. However, recent direct recordings of bipolar cells suggest that such period adaptation is not present [10]. The other model claims that having dual pathways of ON and OFF bipolar cells are enough to reproduce most of the aspects of the phenomena observed in experiments [7]. However, the model only reproduces the shift of the onset of the burst, and not a shift in the peak of the burst, which has the critical predictive latency [8].

Direct model reduction of the deep CNN in Fig. 1A using the methods of section 2 yields for the first time an interpretable model that can reproduce all salient features of the OSR response, comprised of *three* important feedforward pathways that combine one OFF temporal filter with two ON temporal filters (Fig. 2(A-1)). One ON filter has an earlier peak with a longer tail, while the other has a later peak with a shorter tail. The filter with an earlier peak and longer tail yields stronger activity for higher frequency flash trains, thereby explaining the shift of the burst response to earlier times for higher frequency trains (Fig. 2C).

Thus, our systematic model reduction approach yields a new model of the OSR that cures important inadequacies of prior models. Moreover, it yields a new, experimentally testable scientific hypothesis that the OSR is an emergent property of *three* bipolar cell pathways with specific and diverse temporal filtering properties.

# 3  Discussion

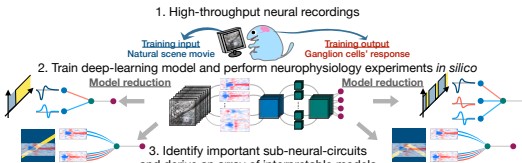

Figure 3: **A unified framework to reveal computational structure in the brain.** We outlined an automated procedure to go from large-scale neural recordings to mechanistic insights and scientific hypotheses through deep learning and model reduction.

In summary, in the case of the retina, we have shown that complex CNN models obtained via machine learning can not only mimic sensory responses to rich natural scene stimuli, but also can serve as a powerful and automatic mechanism for generating valid scientific hypotheses about computational mechanisms in the brain, when combined with our proposed model reduction methods (Fig. 3). Applying this approach to the retina yields conceptual insights into how a *single* model consisting of multiple nonlinear pathways with diverse spatiotemporal filtering properties can explain decades of painstaking physiological studies of the retina. Overall the success of this deep learning approach to scientific inquiry in the retina, which was itself not at all *a priori* obvious before this work, encourages future studies to explore this approach deeper in the brain.

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
