# OpenReview forum: "Revealing computational mechanisms of retinal prediction via model reduction"
_NeurIPS.cc/2019/Workshop/Neuro_AI — Real Neurons & Hidden Units @ NeurIPS 2019 Poster_

### Official Review · AnonReviewer1 · 2019-09-26
**Potentially exciting and widely applicable method; paper could do with more information and less salesmanship**

**Clarity:** 4

**Comment:**

-- A great deal of the text, in both the abstract and manuscript, is given over to bombastic excitement about the power and virtues of the method. I understand the desire to communicate the potential of a new method, but this would be a more informative, self-contained piece of work if more space and attention were dedicated to describing the methods involved in the proof-of-principle study reported.

**Category:**

AI->Neuro

**Clarity Comment:**

-- Text is written articulately, figures are exceptionally rich.

-- Although the CNN used appears to be that from Maheswaranathan et al. (2018), this would be a more self-contained submission if it included some of the details about that model and its training data. For example, there is nothing in the present submission which even specifies which species' retina the CNN is a model of (aside from a non-sequitur mention of salamanders in the text, and an odd-looking creature in Figure 3). Nor is there any detail about the image stimuli to which neural responses were recorded, even though the particular spatiotemporal structure of the stimuli seems to be critical to the model reduction method.

**Evaluation:**

5: Excellent

**Importance:**

4: Very important

**Importance Comment:**

-- Development of new methods to distill DCNN computational strategies is crucial. The present method is interesting and appears to generate testable novel hypotheses.

**Intersection:**

5: Outstanding

**Intersection Comment:**

-- Development of model summarisation and interpretation methods is of critical importance to making AI-based systems useful models in neuroscience.

**Rigor Comment:**

-- Appears rigorous. I leave it to reviewers with more expertise in pre-existing model reduction techniques to evaluate the novelty.

**Technical Rigor:**

4: Very convincing

---

### Official Review · AnonReviewer3 · 2019-09-26
**Potentially significant results for neural network model reduction moderately obscured by unclear writing**

**Clarity:** 2

**Comment:**

This work seeks to take a significant step in a very interesting and important direction, but issues with clarity make it hard to deduce how successful they are in making this step. The discovery of a new circuit mechanism that implements omitted stimulus response more robustly feels like a very significant contribution, although my lack of familiarity with this body of work makes it hard for me to judge with confidence.

**Category:**

Common question to both AI & Neuro

**Clarity Comment:**

The clarity of the work suffers both from missing technical details (see Rigor Comment) as well as inadequate interpretation of the results. Since I've already addressed the former, here I'll focus on the latter.

First, it isn't clear to me how novel equation (1) is. This equation, as far as I can tell, charts a way to making an optimal filter that takes in hidden unit activations before the nonlinearity is applied and outputs a response r(t) that is close to (or exactly equal to) the true response. Since the topic of building optimal filters is a rich and well-explored one, it is essential to contextualize equation (1) within this body of work.

Second, it is difficult to interpret how to think of the "effective weights" they derive, and the resulting effective filters. In particular, the curves that are plotted in Figure 2, for instance in A-1, aren't very well explained -- are these the preactivations of the hidden units, or the filters applied to these preactivations? Why is it that the curves found in the reduced "three hidden unit" model in Figure A-1 aren't found among the filters in the "eight hidden unit" model of figure A-1?

Third, it isn't clear to me how the reduced "filter model" they derive is mapped back into a CNN framework. The paper claims to provide a method to "algorithmically extract computational mechanisms, and consequently conceptual insights, from deep CNN models" by extracting "a simple, reduced, minimal subnetwork that is most important in generating a complex CNN’s response to any given stimulus." However, the "subnetwork" does not appear to me to actually be a subnetwork of the CNN, but rather a different, simpler network model that only overlaps with the CNN at the first layer. Can we really be sure that the CNN is implementing the filters as derived in the reduced model? Validation on a held-out test set may be needed to test this hypothesis.

**Evaluation:**

3: Good

**Importance:**

4: Very important

**Importance Comment:**

This work advances previous work that introduced a three layer convolutional neural network (CNN) model that both reproduces retinal ganglion cell responses with high accuracy and whose hidden neurons are well correlated with retinal interneurons, by elucidating the computations that are happening between input and output neurons in both the CNN and retina recordings. Issues with clarity make it hard to deduce how successful they are in achieving their stated goals.

**Intersection:**

4: High

**Intersection Comment:**

The paper seeks to shed light both on what is going on inside both artificial as well as biological neural networks. Their finding of a new neural circuit mechanism that implements omitted stimulus response more robustly certainly sheds light on the latter, but I don't feel like they made a very clear case when it comes to the former.

**Rigor Comment:**

The authors introduce a decomposition of the output firing rate into a dynamically weighted sum of the pre-nonlinearity activations of hidden units. This derivation is hampered by an important symbol not being defined (the caligraphic F). Perhaps in equation (1) the caligraphic F was actually supposed to be a caligraphic A. It's not stated if this decomposition is unique. The derivation seems to follow from a use of the Fundamental Theorem of Calculus followed by the chain rule applied to the caligraphic F, but they seem to use that this F evaluated at zero input stimulus is zero, and it isn't clear why this would be the case with a nonzero bias.

The dynamically weighted sum is truncated based on the magnitude of the terms. Since these terms depend on time t, it isn't clear how this magnitude is taken.

While the introduction to Section 2 describes the method as "we first carve out important sub-circuits using modern attribution methods, and then reduce dimensionality by exploiting spatial invariances...", in their actual approach it seems like they exploit spatial invariances to reduce dimensionality before carving out the sub-circuits.

One important comparison to have made I think is with a model that builds filters based off of the input stimuli themselves rather than the hidden unit preactivations. How useful is doing the latter compared to the former?

**Technical Rigor:**

2: Marginally convincing

---

### Official Review · AnonReviewer2 · 2019-09-27
**An interesting case study in model reduction with potential broader applications**

**Clarity:** 3

**Category:**

AI->Neuro

**Clarity Comment:**

The flow/high-level organization of the paper works well. Explanations are mostly complete, though some details are missing. e.g. what was the nonlinearity used in the model CNN? Also, do the CNN layers correspond to cell populations, and if so, why is it reasonable to collapse the time dimension after the first layer?

**Evaluation:**

3: Good

**Importance:**

3: Important

**Importance Comment:**

The authors state three high-level improvements they want to make to CNN-based models of neural systems:

1 & 2) Capturing computational mechanisms and extracting conceptual insights. Operationally, I'm not quite sure how these are different, so, to me this goal is roughly "be explainable", and progress towards it could be measured e.g. in MDLs.

3) Suggest testable hypotheses.

I agree these are good goals, and I think some progress is made, but that progress seems somewhat limited in scope.

**Intersection:**

5: Outstanding

**Intersection Comment:**

I believe this paper is addressing questions that many of the workshop attendees will find interesting.

**Rigor Comment:**

The technical aspects of the paper seem correct, though I have some higher-level conceptual concerns.

1) If I understand correctly, attribution is computed only for a single OSR stimulus video. Is the attribution analysis stable for different stimulus frequencies? If not, is it really an explanation of the OSR?

2) I agree with a concern raised by reviewer 3: It's difficult to see a 1-layer network as a "mechanistic explanation" of a 3-layer network.

**Technical Rigor:**

2: Marginally convincing

---

### Decision · Program_Chairs · 2019-10-02

Accept (Poster)